# C1q Complement/Tumor Necrosis Factor-Associated Proteins in Cardiovascular Disease and COVID-19

**DOI:** 10.3390/proteomes9010012

**Published:** 2021-03-01

**Authors:** Yaoli Xie, Zhijun Meng, Jia Gao, Caihong Liu, Jing Wang, Rui Guo, Jianli Zhao, Bernard Lopez, Theodore Christopher, Daniel Lee, Xinliang Ma, Yajing Wang

**Affiliations:** 1Department of Emergency Medicine, Thomas Jefferson University, Philadelphia, PA 19107, USA; Dr_xieyaoli@outlook.com (Y.X.); zhijun.meng@jefferson.edu (Z.M.); jianli.zhao@jefferson.edu (J.Z.); bernard.lopez@jefferson.edu (B.L.); Theodore.christopher@jefferson.edu (T.C.); Tuh11486@temple.edu (D.L.); Xinliang.Ma@jefferson.edu (X.M.); 2Department of Physiology, Shanxi Medical University, Taiyuan 030001, China; gaojiadoctor@outlook.com (J.G.); liucaihongwei@outlook.com (C.L.); first20050903@163.com (J.W.); rui.guo@sxmu.edu.cn (R.G.)

**Keywords:** CTRPs, cardiovascular disease, COVID-19, obesity

## Abstract

With continually improving treatment strategies and patient care, the overall mortality of cardiovascular disease (CVD) has been significantly reduced. However, this success is a double-edged sword, as many patients who survive cardiovascular complications will progress towards a chronic disorder over time. A family of adiponectin paralogs designated as C1q complement/tumor necrosis factor (TNF)-associated proteins (CTRPs) has been found to play a role in the development of CVD. CTRPs, which are comprised of 15 members, CTRP1 to CTRP15, are secreted from different organs/tissues and exhibit diverse functions, have attracted increasing attention because of their roles in maintaining inner homeostasis by regulating metabolism, inflammation, and immune surveillance. In particular, studies indicate that CTRPs participate in the progression of CVD, influencing its prognosis. This review aims to improve understanding of the role of CTRPs in the cardiovascular system by analyzing current knowledge. In particular, we examine the association of CTRPs with endothelial cell dysfunction, inflammation, and diabetes, which are the basis for development of CVD. Additionally, the recently emerged novel coronavirus (COVID-19), officially known as severe acute respiratory syndrome-coronavirus-2 (SARS-CoV-2), has been found to trigger severe cardiovascular injury in some patients, and evidence indicates that the mortality of COVID-19 is much higher in patients with CVD than without CVD. Understanding the relationship of CTRPs and the SARS-CoV-2-related damage to the cardiovascular system, as well as the potential mechanisms, will achieve a profound insight into a therapeutic strategy to effectively control CVD and reduce the mortality rate.

## 1. Introduction

Cardiovascular disease (CVD) is the leading global cause of death, accounting for 17.3 million deaths per year [1]. With the increase in the number of people with diabetes, the morbidity and mortality of CVD are excessively accelerated [2,3]. 

A comprehensive understanding of the relationship between CVD and its risk factors is of great interest in the research community. Recently, a family of C1q complement/tumor necrosis factor (TNF)-associated proteins (CTRPs) has received attention due to their newly discovered role in the cardiovascular system. CTRPs possess broad distribution, participate in multiple aspects of metabolism, and potentiate regulation of homeostasis, and hold potential as diagnostic or therapeutic targets of obesity-related metabolic disorders, including CVD. Unraveling the signaling pathways downstream of CTRP family members will facilitate new insights into therapeutic strategies for CVD.

CTRPs are also of interest in relation to COVID-19, the global pandemic evolving in real time. A growing number of clinical reports indicate that obesity/diabetes is a risk factor for COVID-19 severity in CVD, [4] and patients with cardiac injury and COVID-19 have adverse prognoses. The relationship between of CTRPs and heart-related symptoms in COVID-19, including the effects of the disrupted adipokines on inner homeostasis of cardiovascular system, is of interest.

In this review, we focus on the pathophysiologic roles of CTRPs with cardiovascular disease and summarize the relationship between cardiovascular disease, COVID-19, and diabetes.

## 2. What Are CTRPs and How Are They Related to CVD?

The CTRP superfamily, originally introduced by Harvey Lodish and colleagues, describes a new family of secreted proteins [5,6]. The CTRP superfamily is a paralog of adiponectin (APN), composed of CTRP1–CTRP15, which share a common structural domain with APN [5]. CTRPs are composed of four distinct domains, comprising an N-terminal signal peptide, a short variable domain, a collagen-like domain, and a C-terminal C1q-like globular domain. C1q forms trimers composed of A, B, and C chains [7]. (Figure 1) The globular domain is important for its biological function. The C-terminal region of the tumor necrosis factor (TNF) homology domain (THD), which is similar to that of the globular C1q (gC1q) domain, is a typical feature of members of the TNF family. The C1q and TNF family proteins have similar gene structures: their gC1q or THD domains are each encoded within one exon, while introns are restricted to respective stalk regions [7,8]. CTRPs are secreted by different viscera and tissues, including the adipose tissue, heart, and liver [9]. These adipokines play important roles in obesity, diabetes, and cardiovascular disease. According to current research, CTRP1, CTRP3, CTRP5, CTRP6, CTRP9, CTRP12, CTRP13, and CTRP15 are related to CVD. These proteins influence the progression of CVD by regulating inflammatory responses, endothelial function, metabolic dysfunction, myocardial cell apoptosis, and fibrosis [10,11].

## 3. CTRPs and Vascular Diseases

CTRPs have been reported to play important roles in maintaining homeostasis of the vascular system against endothelial dysfunction through regulating inflammatory responses and correcting metabolic imbalance.

### 3.1. CTRPs and Endothelial Cell Dysfunction

Endothelial dysfunction, the earliest alteration in vascular pathology, plays a critical role in atherosclerosis development. Several CTRPs are involved in regulating endothelial pathophysiological progression.

CTRP3 can efficiently inhibit the inflammatory response and endothelial dysfunction induced by oxidized low-density lipoproteins (oxLDLs) in mouse aortic endothelial cells by activating the PI3K/Akt/eNOS pathway [12]. Inflammation plays a key part in atherosclerosis, so these results suggest that overexpressed CTRP3 may play a role in a novel approach for preventing inflammation and endothelial dysfunction and inhibiting atherosclerosis.

CTRP5 promotes early-stage atherosclerosis through two synergistic mechanisms: facilitating entry of circulating LDLs into the subendothelial space via transcytosis and inducing oxidation of LDLs by endothelial cells (ECs) [13]. Furthermore, CTRP5 exerts these effect via upregulating 12/15-lipoxygenase (LOX) expression through STAT6 signaling, thus facilitating transcytosis of low-density lipoproteins (LDLs) across endothelial monolayers and inducing LDL oxidation [13].

Among CTRPs, CTRP9 exerts by far the highest expression in the heart and can also be found in serum and adipose tissue [14,15,16]. CTRP9 is also the closest paralog of APN and has the highest amino acid sequence similarity (54%) at the globular domain to APN [17]. CTRP9 attenuates TNFa-induced NF-kB activation in vascular endothelial cells, thereby downregulating the NF-kB-dependent gene expression of cell adhesion and inflammatory molecules (e.g., ICAM-1, VCAM-1, MCP-1). The process is mediated by AMPK activation in endothelial cells [18]. The protective effects of CTRP9 on endothelial oxidative damage are likely associated with enhanced mitochondrial biogenesis through the sirtuin-1 (SIRT1)-dependent proliferator-activated receptor-coactivator-1α (PGC-1α) pathway, which is AdipoR1-dependent [19]. CTRP9 induces endothelial NO production to elicit vasorelaxation in an AMPK-dependent manner, and AdipoR1 acts as a receptor of CTRP9 [20]. CTRP9 therefore provides a protective function in endothelial cells (Figure 2).

For the therapeutic prevention of vascular disease, CTRP3 may be a novel target for preventing atherosclerosis through anti-inflammation while CTRP5 and CTRP9 may be more involved in the metabolic process than inflammatory response. However, although many of the mechanisms of CTRPs have been discovered, the direct receptors of CTRPs are still unclear. It is a great pursuit to explore the detection of these mediators on the cell surface, which will contribute to the prospects of clinical therapeutic target development.

### 3.2. CTRPs and Atherosclerosis

Atherosclerosis is a chronic inflammatory disease of the arterial wall and the primary underlying cause of CVD. The origin of atherosclerosis is related to lipid metabolism alterations, chronic inflammation, and oxidative stress. Many studies have demonstrated that several CTRPs are involved in regulating the pathophysiological progression of atherosclerosis [21,22]. In particular, current research indicates that various CTRPs are involved in the development and progression of atherosclerosis by regulating inflammation response, lipid metabolism, and vascular smooth muscle cell (VSMC) proliferation [23,24,25].

CTRP1 reduces VSMC growth through the cAMP-dependent pathway to prevent the development of pathological vascular remodeling [26]. Compared to a control group, patients with severe coronary artery disease (CAD) had higher levels of CTRP1 in sera, coronary endarterectomy samples, atherosclerotic plaques, and peripheral blood mononuclear cells (PBMCs) [27]. CTRP1 increases the expression of adhesion molecules and inflammatory cytokine through the p38 MAPK/NF-kB pathway, thereby promoting leucocyte adhesion to endothelial cells in vitro and in vivo, suggesting that CTRP1 is an adipokine contributing to atherogenesis [27]. However, which CTRP1-expressing cell type contributes to promoting atherosclerosis and whether treatments targeting CTRP1 will prevent further progression of established atherosclerosis and induce plaque regression have not been confirmed. Further research is required.

In addition to CTRP1, CTRP3, CTRP6, and CTRP12, are involved in the regulation of metabolism. They balance glucose levels by suppressing hepatic gluconeogenesis and glucose output [28]. CTRP3 acts as lipopolysaccharide (LPS) antagonist of the adipose tissue to block the proinflammatory activation of adipocytes and monocytes. It does so by inhibiting three basic and common proinflammatory pathways: it inhibits the release of chemokines in monocytes and adipocytes; it inhibits monocyte chemoattractant protein-1 release in adipocytes; it inhibits the binding of LPS to its receptor, TLR4/MD-2 [29]. CTRP6 mediates fatty acid oxidation via the AMPK-ACC pathway [30]. CTRP12, an adipokine with antidiabetic actions, preferentially acts on adipose tissue and liver to control whole body glucose metabolism [31]. CTRP12 have been found to be related to several CAD risk factors, including BMI, inflammatory cytokines, insulin resistance, high-density lipoprotein-cholesterol, and adiponectin [32]. This result suggests a possible link between CTRP12 and pathogenic mechanisms of atherosclerosis [32]. Endogenous CTRP12 protects against the development of pathological vascular remodeling by attenuating macrophage inflammatory responses and VSMC proliferation through transforming the growth factor-β receptor II (TGF-βRII)/Smad2-dependent pathway in an established mouse model of vascular injury [33]. In addition, the TGF-β1/Smad signaling pathway is responsible for vascular fibrosis during the development of atherosclerosis [34]. Thus, it is necessary to further investigate whether CTRP12 promotes vascular disease by increasing arterial fibrosis.

In regard to CTRP regulation in inflammatory response, in addition to CTRP1, CTRP5 promotes inflammation, proliferation, and migration in vascular smooth muscle cells through activation of Notch1, TGF-β, and hedgehog signaling pathways [35]. CTRP6 stimulates expression of IL-10, an anti-inflammatory cytokine, in macrophages through activating extracellular signal-regulated kinase1/2 (ERK1/2) [36]. In addition, CTRP6 inhibits platelet-derived growth factor-BB (PDGF-BB)-induced VSMC proliferation and migration, partially, via suppression of the PI3K/Akt/mTOR signaling pathway [37]. The above results suggest that CTRP5 and CTRP6 may be a potential target for the treatment of atherosclerosis through regulating inflammatory responses and the proliferation and migration of VSMC. Additionally, there are reports suggesting that serum CTRP5 levels were higher in in-stent restenosis (ISR) patients than in non-ISR patients after drug-eluting stent (DES)-based percutaneous coronary intervention (PCI) [35].

Although CTRP9 has been called cardiokine due to its high levels in the cardiac system, it alters the components of carotid plaque and decreases inflammatory cytokines in atherosclerotic plaque both in vivo and in vitro. As a result, CTRP9 may enhance carotid plaque stability and play an anti-inflammation role against atherosclerosis [38]. CTRP9 reduces the inflammatory response to oxidized low-density lipoprotein in cultured macrophages via an AMPK-dependent mechanism [39]. CTRP9 prevents VSMC proliferation and neointimal thickening following mechanical arterial injury [23]. Of importance, the antiproliferative effects of CTRP9 are attributed to reduced ERK activation that is involved in the cAMP-PKA regulatory axis [23]. CTRP9 slows the pathological progression of early atherosclerosis by promoting cholesterol efflux to reduce the formation of foam cells, and the AMPK/mTOR autophagy signaling pathway is a response to regulate this process [40]. CTRP9 has also shown atheroprotective function via the CTRP9-AMPK-NLR family pyrin domain containing 3 (NLRP3) inflammasome pathway [41]. These studies indicate CTRP9 may play an antiatherogenic role. CTRP9 possesses vascular protective effects and involves multiple signaling pathways to modulate the inflammatory response in the cardiovascular system. However, it is necessary to use specific endothelial cell transgenic animal models in future studies the dissecting the role endogenous CTRP9 plays in vascular disease.

The role of CTRP13 in atherosclerosis is different from others. It focuses on the immunocytes other than the direct effect on cardiovascular cells. CTRP13 exerts a protective effect in atherosclerotic plaque development through inhibition of macrophage lipid uptake and preservation of the migration of macrophages [42,43]. It promotes autophagy in macrophages and accelerates autophagy-lysosome-dependent degradation of CD36, thus inhibiting macrophage lipid uptake and foam-cell migration [44]. CTRP13 hydrolyzes cholesterol droplets stored in macrophages, inhibits intracellular influx of cholesterol, and promotes cholesterol efflux, thus inhibiting the formation of foam cells and decelerating progression of atherosclerosis [45,46]. CTRP13 has also showed its metabolic regulatory function. Although it is an adipokine that promotes glucose uptake in adipocytes, myotubes, and hepatocytes via activation of the AMPK signaling pathway, CTRP13 also ameliorates lipid-induced insulin resistance in hepatocytes by suppressing the SAPK/JNK stress signaling that impairs the insulin signaling pathway. Further, CTRP13 reduces glucose output in hepatocytes by inhibiting the mRNA expression of gluconeogenic enzymes, glucose-6-phosphatase, and the cytosolic form of phosphoenolpyruvate carboxykinase [47]. Therefore, CTRP13 may be a novel therapeutic approach for attenuating the progression of atherosclerosis via immunoregulation combined with body metabolic rebalance (Figure 3).

### 3.3. CTRPs and Diabetic Vascular Disease

Diabetes is undoubtably a risk factor that accelerates CVD. Among diabetic vascular complications, diabetic atherosclerosis is a major contributory factor to CVD [44,48]. It has been increasingly recognized that diabetic and obesity states cause chronic low-grade inflammation. The resultant deleterious effects of ambient cytokines, as well as high-glucose levels, further exacerbate the inflammatory response and enhance leukocyte–endothelial interactions, leading to elementary atherosclerotic processes. In light of CTRPs role in the regulation of inflammation and metabolism, CTRPs in diabetic vascular disease cannot be unneglected.

CTRP1 was greatly induced after oxLDL exposure by activating nuclear receptor (peroxisome proliferator-activated receptor) PPAR-r. This inducing of CTRP1 increased the secretion of inflammatory cytokines (e.g., MCP-1, TNF-a, IL-1β), thus promoting the development of atherosclerosis [22]. CTRP9 might be important in the regulation of arterial stiffness in humans based on findings that serum CTRP9 concentration is significantly and positively associated with arterial stiffness in T2DM patients [49]. In T2DM and CAD patients, CTRP9 is positively correlated with BMI, glucose metabolism parameters, inflammatory markers, and adhesion molecules, and is negatively correlated with adiponectin. Increased levels of circulating CTRP9 in individuals with T2DM and CAD suggest a compensatory response to insulin resistance, inflammatory milieu, and endothelial dysfunction [50] (Figure 4). Schmid and colleagues illustrated that CTRP-5 might be an adipokine, which has a counter-regulatory connection with its family member, CTRP-3 [51]. Moreover, earlier studies have shown that serum CTRP5 levels are significantly higher in obese/diabetic animals, and the expression and secretion of CTRP5 correlates negatively with mtDNA content in myocytes [52]. This study also demonstrated that gCTRP5 shows similar biological activities to adiponectin, such as activating AMPK and increasing glucose uptake and fatty acid oxidation [52].

Although the roles of CTRPs in the modulation of diabetic vascular disease are indirect, the direct roles in favor of decelerated diabetic vascular disease are still under exploration until the publication in 2020 [21], with the aim of demonstrating how globular CTRP5 (gCTRP5) directly influences on diabetic vascular disease. gCTRP5 is accumulated in diabetic circulatory systems and appears to contribute to diabetic vascular EC dysfunction through Nox1-mediated mitochondrial apoptosis. gCTRP is one of the signaling molecules (along with HFD and HGHL) that commonly activates expression of Nox1, which is implicated in the pathogenesis of cardiovascular diseases. Research reasoned that gCTRP5 activates the mitochondrial apoptotic signal of EC in diabetes, which is blocked by the silencing Nox1 gene [21]. This study’s authors suggest that interventions blocking gCTRP5 may protect diabetic EC function, ultimately protecting against diabetic cardiovascular complications [21]. Pharmacological interventions targeting CTRP5 or its related signaling may provide promising therapeutic avenues to attenuate the development of atherosclerosis or diabetic EC dysfunction and cardiovascular complications. To date, the role of CTRP5 in cardiac diseases is still unclear, but the above studies provide evidence for its potential direct role in diabetic various cardiac diseases and so on.

The relationship of CTRPs with diabetes is more complex than expected. According to clinical research data, CTRPs are responsible for the majority of the changes that occur at the different levels of diabetes, including insulin resistance. Whether CTRPs can serve as diagnostic or prognostic markers and a full view of the role of CTRPs in diabetes need to be further investigated and discussed.

## 4. CTRPs and Cardiac Diseases

Growing evidence has confirmed that CTRPs exerts crucial effects on cardioprotection aspects through anti-inflammation, antiapoptosis, antifibrosis, and proangiogenesis.

### 4.1. CTRPs and Heart Failure

Cardiac hypertrophy is an adaptive response to maintain cardiac function, but it ultimately becomes mainly maladaptive and leads to heart failure (HF). However, accumulated data provided evidence that CTRPs are highly associated with cardiometabolics, inflammatory response, and immunoregulatory function [53,54,55].

CTRP3 expression is upregulated in hypertrophic and failing hearts in murine models [56]. CTRP3 also promotes pressure overload-induced cardiac hypertrophy via activation of the transforming growth factor β-activated kinase 1-c-Jun N-terminal kinase (TAK1-JNK) axis [56]. CTRP9 is upregulated in hypertrophic heart disease and induces cardiomyocyte hypertrophy and cardiac dysfunction by activating ERK5 after transverse aortic constriction (TAC) in mice [57]. Furthermore, this study identified GATA4 as a downstream target of CTRP9-ERK5 [57]. GATA4 could account for increased hypertrophy during CTRP9-ERK5 signaling, but likely not for cardiac dysfunction. Thus, additional downstream targets of the CTRP9-ERK5 probably contribute and will need to be identified. These observations advance our understanding of relationship between CTRPs and heart failure (Figure 5).

### 4.2. CTRPs and Ischemic Cardiac Disease

The main characteristic of CTRPs are metabolic and inflammatory regulation. Accumulated data have demonstrated metabolic-related dysfunction was elicited during ischemic heart disease. Ischemic cardiac disease is a type of disease given to heart problems caused by narrowed heart arteries that lead to myocardiac ischemia (MI). A variety of links have been made between CTRPs and ischemic cardiac disease [58]. The involved mechanisms has been clarified in many aspects.

CTRP1 protects against myocardial ischemic injury by reducing apoptosis and inflammatory response through activation of the sphingosine-1-phosphate (S1P)/cAMP signaling pathways in cardiomyocytes, suggesting that CTRP1 plays a protective role in ischemic heart disease [59]. Thus, CTRP1 plays an opposite role in vascular diseases and cardiac diseases, and the underlying mechanism involved needs to be further studied. Meanwhile, research indicates CTRP3 has a protective effect on the cardiovascular system, involving multiple signaling pathways. The expression and production of CTRP3 are significantly reduced post-MI, [60] and replenishment of CTRP3 attenuates post-MI pathologic remodeling, including reducing heart size and cardiomyocyte apoptosis, increasing cardiomyocyte survival/regeneration, attenuating remote area interstitial fibrosis, as well as enhancing infarct border zone revascularization [60]. Furthermore, by promoting cardiomyocyte–endothelial cell communication involving Akt-HIF1α-VEGF signaling, CTRP3 exerts an angiogenic effect [60]. CTRP3 further exerts a cardiac antifibrotic effect post-MI by inhibiting myofibroblast differentiation and the subsequent extracellular matrix production. AMPK is required for the protective effect of CTRP3 against TGF-β1-induced profibrotic response. The effect of CTRP3 may be achieved by targeting the Smad3 signaling pathway [61] (Figure 3). These findings provide further understanding of the molecular mechanisms of CTRP3 in cardiac protection and provide new insights into therapeutic targets for cardiac remodeling. Perhaps preventing post-MI CTRP3 inhibition or CTRP3 supplementation can act as a promising therapeutic approach for creating stable and functional vessels post-MI, restoring cardiac function, and improving the heart failure phenotype.

CTRP6 is a cardioprotective adipokine that ameliorates ventricle remodeling post-MI, including inhibiting myofibroblast differentiation, extracellular matrix (ECM) production, and cardiac fibroblast (CF) migration [62]. AMPK and Akt activation contribute to the protective effect of CTRP6 against TGF-β1-induced fibrotic response by targeting the Smad-independent myocardin-related transcription factor-A (RhoA/MRTF-A) signaling pathway [62]. At present, there are no reports on the effects of CTRP6 on cardiomyocyte apoptosis and angiogenesis post-MI. Further study is needed on the effects of CTRP6 and its potential mechanisms. Recently, research has found that CTRP12 also ameliorates pathological remodeling of myocardium after MI by reducing myocardial inflammatory response and apoptosis in vivo [63]. Furthermore, CTRP12 reduces inflammatory response and apoptosis of cardiomyocytes through the PI3 kinase/Akt signaling pathway [63]. CTRP15, which is expressed abundantly in skeletal muscle and to a lesser extent in the lung, eye, smooth muscle, heart, and brain, exerts multiple biological effects, including promoting lipid metabolism in hepatocytes and adipocytes, suppressing autophagy in the liver, and modulating erythropoiesis [64,65,66,67,68]. CTRP15 is also known as myonectin [69]. Studies have found that CTRP15 inhibits the fibrotic response through attenuating myofibroblast differentiation and expression of profibrotic molecules on pressure overload-induced cardiac remodeling. The beneficial effects of CTRP15 on the TGF-β1-induced fibrotic response is through the IR/IRS-1/PI3K/Akt pathway. Smad3 also participates in this CTRP5′s role [70]. CTRP15 suppresses cardiomyocyte apoptosis and macrophage inflammatory response through the S1P-dependent activation of the cAMP/Akt pathway in the heart, thereby ameliorating acute myocardial ischemic injury [71]. The receptor involved in mediating these signaling pathways in cardiovascular tissues is not known and needs to be clarified in future studies (Figure 6).

CTRP9 is the only cardiokine which is widely accepted and recognized in the CTRP family. CTRP9 is the most abundantly expressed adipokine in the heart, exceeding local APN expression more than 100-fold, with local cardiac CTRP9 levels exceeding plasma CTRP9 levels more than 2-fold [14]. It undergoes proteolytic cleavage to generate gCTRP9, its dominant circulatory and biologically active isoform [16]. Serum CTRP9 is an independent protective factor of CAD [72]. Serum CTRP9 decreases significantly, and the protein and mRNA expressions of CTRP9 in epicardial adipose tissue (EAT) are reduced markedly in CAD patients compared to non-CAD patients [72]. Although the clinical significance has been identified, basic studies clarified why CTRP9 has strong effect on the cardiac function with the following properties. Serum CTRP9 is negatively associated with traditional risk factors of cardiovascular diseases and some inflammatory factors, but it is positively associated with serum APN and high-density lipoprotein-cholesterol (HDL-C) [72]. These associations suggest that circulating and coronary CTRP9 play important roles in the inflammation that occurs in CAD [57]. CTRP9 treatment significantly decreases matrix metalloproteinase2/matrix metalloproteinase 9 (MMP2/MMP9) activity and TGF-β1 production, the two most significant mechanisms contributing to post-MI fibrosis [73]. CTRP9 treatment also dramatically attenuates apoptotic cell death and significantly suppresses interstitial fibrosis [73]. Collectively, CTRP9 prevents left ventricle (LV) remodeling by reducing apoptosis and fibrosis, and this occurs largely via a PKA-dependent pathway [73]. Studies in chronic intermittent hypoxia (MI+CIH) animals found that CTRP9 attenuates interstitial fibrosis, improves cardiac function, and enhances survival rate via inhibiting TGF-β/Smad and Wnt/β-catenin pathways [74]. MI+CIH upregulates the expression of miR-214-3p, which is the target of CTRP9 gene [74]. Altogether, MI+CIH suppresses cardiac CTRP9 expression by upregulating miR-214-3p, and exacerbating post-MI remodeling [74]. These findings provide evidence that CTRP9 and its related signaling maybe a novel therapeutic target in improving cardiac function and alleviating the heart failure (HF) phenotype in MI patients with obstructive sleep apnea (OSA). CTRP9 contributes to cardiac hypertrophy and failure during pressure overload in part through activating the ERK5-GATA4 pathway [57]. In the above research regarding the role of CTRP9 on remodeling post-MI, the CTRP9 was administered before or shortly after ischemia and alleviated adverse cardiac remodeling. In general, CTRP9 plays an crucial role in attenuating atrial inflammation, fibrosis, and atrial fibrillation post-MI in the following ways: it markedly downregulates inflammatory factors, interleukin-1β and interleukin-6, and upregulates interleukin-10 in 3 days post-MI to ameliorate macrophage infiltration and inflammatory responses; it reduces the expressions of collagen types I and III, a-SMA, and transforming growth factor β1 in 7 days post-MI by depressing the Toll-like receptor 4/nuclear factor-kβ and Smad2/3 signaling pathway, thus playing an antifibrotic role in the left atrium; CTRP9 ameliorates vulnerability to atrial fibrillation in post-MI rats [75]; it can epigenetically modulate microRNAs to adjust the genes expression. Nevertheless, it is unclear whether CTRP9 can reverse pathological immunoresponses and remodeling that have already developed. Clarification of this issue will require screening out a more delayed interval for CTRP9 administration.

### 4.3. CTRPs and Diabetic Cardiomyopathy

Diabetic cardiomyopathy (DC) is a diabetes mellitus (DM)-induced pathophysiological condition that can result in HF. DC is characterized by myocardial fibrosis, dysfunctional remodeling, and associated diastolic and systolic dysfunctions, and eventually HF [76]. To date, there are few studies on the role of CTRPs in DC. CTRP3 in the heart protects against diabetes-related cardiac dysfunction, oxidative damage, inflammation, and cell death in vivo. Moreover, the protective effect of CTRP3 is mediated by activation of AMPKα, and CTRP3 activates AMPKα via the cAMP–EPAC–MEK–LKB1 pathways in vivo [77]. Its downregulation by TNF-α-initiated oxidative PPARγ suppression exacerbates cardiac injury in diabetic hearts [14]. These limited results may provide some theoretical basis for the role of CTRPs in DC. In addition to CTRP9, CTRP1, CTRP4, and CTRP7 are also expressed in the heart at levels significantly greater than APN. However, the impact of these CTRPs on DC are unclear. Future studies should examine the roles of these CTRPs and their underlying mechanisms in DC. (Figure 7)

## 5. CTRPs and the Role in CVD Patients with COVID-19

The outbreak of SARS-Cov-2 (Coronavirus Disease-2019 (COVID-19)) has been claim a public health emergency of international concern [78]. Its effects include injury to the cardiovascular system.

### 5.1. Role of COVID-19 in Cardiovascular Injury

The genome of CoVs is a single stranded positive-sense RNA, and SARS-CoV-2 contains four structural proteins (S, E, M, and N) and sixteen nonstructural proteins (nsp1-16) [79,80,81,82]. Angiotensin-converting enzyme 2 (ACE2) has been identified as a functional receptor for SARS-CoV and SARS- CoV-2 [81,83,84]. Thus, the expression of ACE2 is a critical determinant for the entry of the virus. ACE2 is a membrane-bound aminopeptidase that exerts a key effect in the cardiovascular and immune systems [83,85,86]. Recently, further research has identified ACE2 as highly expressed in pericytes of adult human hearts, [87] and patients with basic heart failure disease exhibited increased expression of ACE2, suggesting an intrinsic susceptibility of the heart to SARS-CoV-2 infection [87,88]. Clinical direct evidence was provided in early studies in China, where 7–20% of patients with COVID-19 were observed with damage to the cardiac system [88,89,90,91,92]. In one study, 5 out of 41 COVID-19 patients showed SARS-COV-2-related myocardial injury, mainly manifested by increased levels of high-sensitivity cardiac troponin I (hs-cTnI) levels (>28 pg/mL) in Wuhan [88]. Therefore, in addition to causing respiratory damage, COVID-19 disease could also damage the cardiovascular system [93,94].

In addition, myocardial injury and heart failure accounted for 40% of deaths in one Wuhan cohort [95]. One study reported a 16.7% incidence of arrhythmia in 138 Chinese COVID-19 patients [89]. Brain-type natriuretic peptide (BNP) levels were also elevated in Washington ICU inpatients [96]. Thus, cardiac injury is prevalent in COVID-19 and apparently impacts prognosis. However, the mechanism of COVID-19 caused cardiac injury is unknown. Although accumulating data reveal elevated inflammatory cytokines levels (including IL-2, IL-7, IL-10, granulocyte colony- stimulating factor (G-CSF), IP-10, MCP1, macrophage inflammatory protein 1α (MIP1α) and TNF) and percentage of CD14^+^CD16^+^ inflammatory monocytes in blood plasma of patients with severe COVID-19, [97] evidence for whether these cause a storm of inflammation in the heart is lacking. The myocardial injury may involve increased cardiac stress due to respiratory failure and hypoxemia; indirect injury due to systemic inflammation; direct myocardial infection by SARS-COV-2; or a combination of all three factors.

### 5.2. Diabetes in CVD Patients with COVID-19

Other than inflammation, a growing number of clinical reports indicate that obesity is a risk factor for COVID-19 severity [4,98] and an adverse prognosis [97]. In COVID-19 infections, one notable feature of this disease is that the high prevalence of obese patients among the most severe cases. Recent evidence indicates that obese states cause chronic low-grade inflammation, which contributes to the development of CVD [5,99]. In obese state, the secretion of proinflammatory adipokines, such as leptin, TNF-α, IL-6, and IL-1β, are upregulated, but anti-inflammatory adipokines are downregulated, especially adiponectin [5,99]. The chronic low-grade inflammation that contributes to the innate immune system might be already in a “primed state” that could promote an hyperinflammatory response [100,101]. Thus, the obese state may amplify the proinflammatory response to SARS-CoV-2 infection and induce more serious cardiovascular damage compared to the nonobese state [100,101]. In addition, two months after discharge from the hospital, 70% of patients with COVID-19 showed cardiac dysfunction and significant cardiac fibrosis. The TGF-β1/Smad3 pathway may be involved in that process [102]. Varga et al. proved the presence of viral elements and accumulation of inflammatory cells within endothelial cells of COVID-19 patients, [103] which suggests that SARS-CoV-2 infection could cause endothelial dysfunction.

### 5.3. CTRPs in Patients with COVID-19

Treatment for COVID-19 should give particular attention to cardiovascular protection, especially in patients who have obesity or diabetes [104]. The mechanism of cardiovascular injury caused by SARS- CoV-2 infection might be related to ACE2 and inflammatory cytokine storms [105,106]. Numerous adipokines exert imbalanced expressions in the obese or diabetes state, which leads to a series of inflammation response and cardiovascular damage. Many studies have demonstrated that adiponectin plays a crucial role in anti-inflammatory and cardiovascular protection [107]; however, adiponectin resistance is present in obesity state and the expression level is downregulated in the diabetes state. CTRPs share a common structural domain with adiponectin and some (e.g., CTRP3, CTRP9, CTRP12) also exhibit anti-inflammatory and cardiovascular protective roles. Controlling the inflammatory response via CTRPs may be a potential target for relieving cardiac injury during treatment for COVID-19. Although current knowledge on CTRPs with COVID-19 is largely unknown, in consideration of CTRPs’ roles in modulating cardiovascular metabolism and preventing inflammatory response, the enhanced understanding and potential use in clinical treatment of cardiaovascular injury of patients with COVID-19 are promising.

## 6. Prospective

Although the discovery of new proteins is driven by cutting-edge techniques including proteomics, the in-depth exploration of proteomics technology still need to broaden the new insight/innovation on protein biological functions by improving the performances of mass spectrometers. The newly discovered CTRP family plays an important role in CVD not only by regulating immuno-inflammation, glucose and lipid metabolism, and vascular endothelial function, but also by reducing cardiomyocyte apoptosis and fibrosis and by ameliorating cardiac function. The CTRP family reveals an exciting avenue for CVD therapeutics. CTRPs and their related signals hold potential to be used as biomarkers or therapeutic interventions against cardiovascular disease, including in patients who have been infected with COVID-19.

## Figures and Tables

**Figure 1 proteomes-09-00012-f001:**
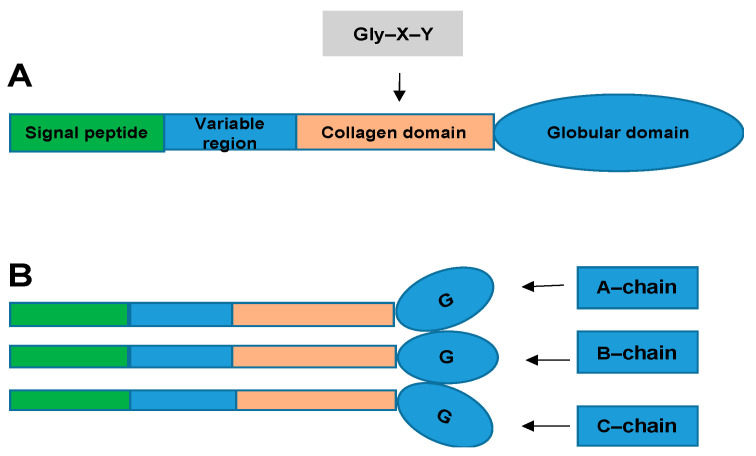
Structural organization of the C1q complement/tumor necrosis factor (TNF)-associated proteins (CTRPs). (**A**): Domain structure of a CTRP monomeric protein. (**B**): Homotrimeric CTRP protein structure. CTRP monomeric proteins form homotrimeric protein structures.

**Figure 2 proteomes-09-00012-f002:**
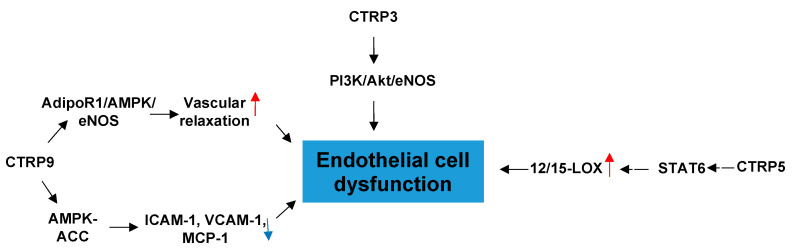
The role of CTRPs in endothelial cell dysfunction and involved mechanisms.

**Figure 3 proteomes-09-00012-f003:**
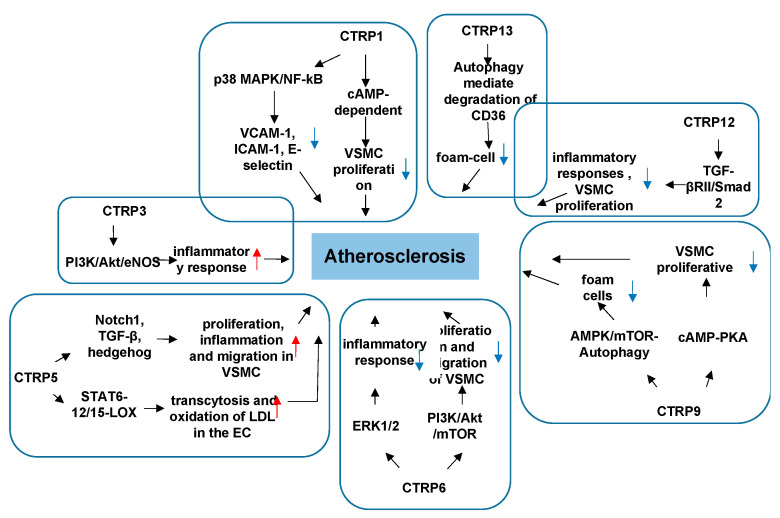
The role of CTRPs in atherosclerosis and involved mechanisms.

**Figure 4 proteomes-09-00012-f004:**
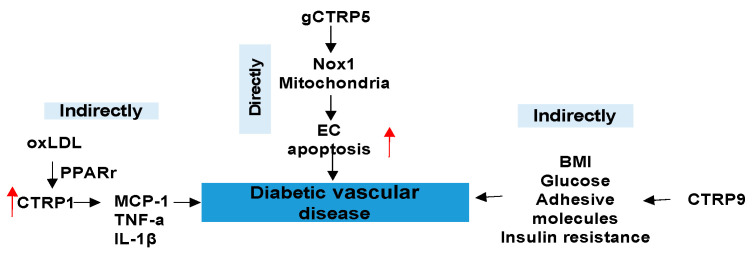
The role of CTRPs in diabetic vascular disease and involved mechanisms.

**Figure 5 proteomes-09-00012-f005:**
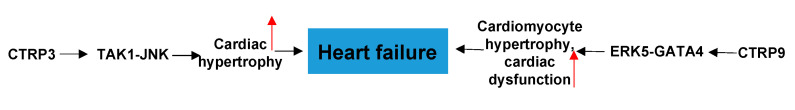
The role of CTRPs in heart failure and involved mechanisms.

**Figure 6 proteomes-09-00012-f006:**
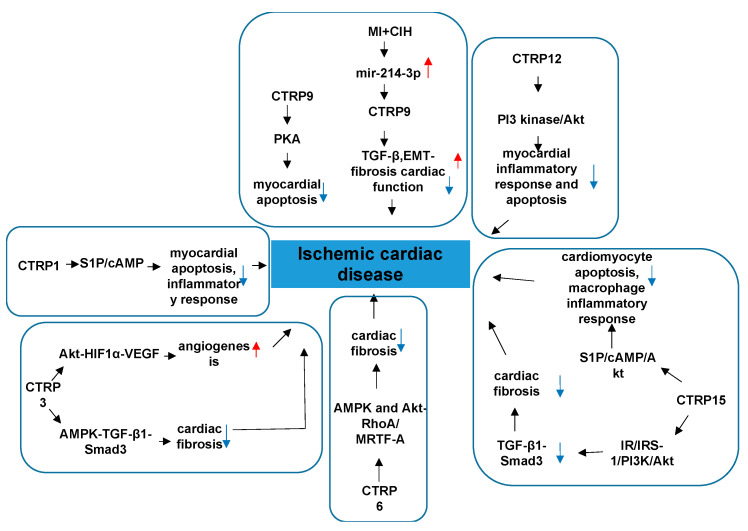
The role of CTRPs in ischemic cardiac disease and involved mechanisms.

**Figure 7 proteomes-09-00012-f007:**
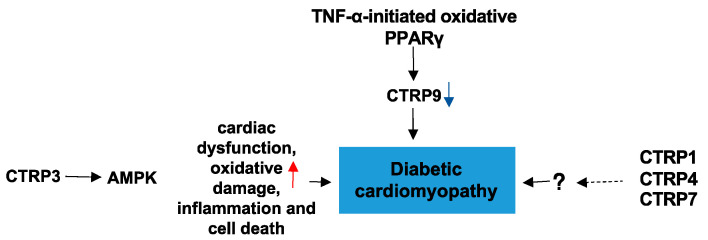
The role of CTRPs in diabetic cardiomyopathy and involved mechanisms.

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
