# Peer review of "C1q Complement/Tumor Necrosis Factor-Associated Proteins in Cardiovascular Disease and COVID-19"

_proteomes, 2021, doi:10.3390/proteomes9010012_

Round 1
Reviewer 1 Report
Dear editor,
I have read the paper entitled with great interest. The review, is nicely crafted and reads well.
To improve readability, the reviewer suggests:
- editing figure 2 by adding CTRP5 actions
There are a few minor spelling mistakes:
- line 86: “CTRP3 maybe acts”
- line 91 “via up-regulates 12/15 (…)”
- line 164 “there are the reports suggested (…)”
- line 248 “CTRPs has majority of the changes that occur (…)”
- line 294 “This ole of CTRP3 may through targeting the Smad3 signaling pathway”
- lines 453 “Although the CTRPs discovery history is tightly linked with proteomics driven discovery. It seems that “
Finally, authors may want to consider acknowledging the following relevant references:
- Sakamoto et al Atheroscler Thromb Vascular Biology 2021; 41(1):542-544
- Tomasoni et al Our J Heart Fail 2020 Nov 12 Nov 12. doi: 10.1002/ejhf.2052. Online ahead of print
- Lombardi et al JAMA Cardiol 2020 Nov 1;5(11):1274-1280. doi: 10.1001/jamacardio.2020.3538
Author Response
We greatly appreciate the reviewer’s positive comments and constructive suggestion, which significantly strengthened our study. Specifically, we have made the following revisions:
Q1. editing figure 2 by adding CTRP5 actions
Response1: We have added CTRP5 actions to the Figure 2.
Q2. spelling mistakes:
- line 86: “CTRP3 maybe acts”
- line 91 “via up-regulates 12/15 (…)”
- line 164 “there are the reports suggested (…)”
- line 248 “CTRPs has majority of the changes that occur (…)”
- line 294 “This ole of CTRP3 may through targeting the Smad3 signaling pathway”
- lines 453 “Although the CTRPs discovery history is tightly linked with proteomics driven discovery. It seems that “
Response2: We thank the reviewer for noting these errors, which has been corrected in the revised manuscript.
Q3. provided relevant literature for us to refer to and cite.
Response3: We have included the provided references in the revised manuscript.

Reviewer 2 Report
Authors have presented the current understanding of C1q complement/tumor necrosis factor-associated proteins (CTRPs) that display differential effects on the regulation of metabolic homeostasis, cardiovascular disease, and COVID-19. The review article is well structured into sections and sub-sections. English is clear and professional. It is within the scope of journal. The review provides a good coverage of the area.
However, it will be beneficial to incorporate some of the interesting studies to provide a comprehensive review. Also, there are studies available in literature that elaborate CTRPs roles in coronary artery disease (CAD) and their exploration as biomarkers. Some suggestions are as follows:
Zhang Y, Liu C, Liu J, Guo R, Yan Z, Liu W, Lau WB, Jiao X, Cao J, Xu K, Jia Y, Ma X, Wang Y. Implications of C1q/TNF-related protein superfamily in patients with coronary artery disease. Sci Rep. 2020 Jan 21;10(1):878. doi: 10.1038/s41598-020-57877-z. PMID: 31965030; PMCID: PMC6972732.
Si Y, Fan W, Sun L. A Review of the Relationship Between CTRP Family and Coronary Artery Disease. Curr Atheroscler Rep. 2020 May 28;22(6):22. doi: 10.1007/s11883-020-00840-0. PMID: 32468164; PMCID: PMC7256102.
Kasher Meron M, Xu S, Glesby MJ, Qi Q, Hanna DB, Anastos K, Kaplan RC, Kizer JR. C1q/TNF-Related Proteins, HIV and HIV-Associated Factors, and Cardiometabolic Phenotypes in Middle-Aged Women. AIDS Res Hum Retroviruses. 2019 Nov/Dec;35(11-12):1054-1064. doi: 10.1089/AID.2019.0099. Epub 2019 Sep 3. PMID: 31359766; PMCID: PMC6862952.
Ni XN, Yan SB, Zhang K, Sai WW, Zhang QY, Ti Y, Wang ZH, Zhang W, Zheng CY, Zhong M. Serum complement C1q level is associated with acute coronary syndrome. Mol Immunol. 2020 Apr;120:130-135. doi: 10.1016/j.molimm.2020.02.012. Epub 2020 Feb 28. PMID: 32120180.
Ilbeigi D, Khoshfetrat M, Afrisham R, Rahimi B, Gorgani-Firuzjaee S. Serum C1q/TNF-Related Protein-2 (CTRP2) Levels are Associated with Coronary Artery Disease. Arch Med Res. 2020 Feb;51(2):167-172. doi: 10.1016/j.arcmed.2020.01.009. Epub 2020 Mar 6. PMID: 32147289.
Fadaei R, Moradi N, Baratchian M, Aghajani H, Malek M, Fazaeli AA, Fallah S. Association of C1q/TNF-Related Protein-3 (CTRP3) and CTRP13 Serum Levels with Coronary Artery Disease in Subjects with and without Type 2 Diabetes Mellitus. PLoS One. 2016 Dec 29;11(12):e0168773. doi: 10.1371/journal.pone.0168773. PMID: 28033351; PMCID: PMC5199067.
The perspective section (page 12, Line 453 onwards) needs to be modified accordingly.
Apart from this, there are some minor issues that are required to be addressed to improve the manuscript.
- Page 3, Line 84: Expand ox-LDL as it is mentioned for the first time in the text. Later use acronym (Page 6, Line 214).
- Page 3, Line 91: The sentence needs reframing for clarity. Suggestion – “... effect via upregulation of…”
- Page 3, Line 96: It should be sequence similarity (54%).
- Page 4, Line 121, 124: The references are required when referring to other studies.
- Page 4, Line 124: Expand VSMC here and later use as acronym (Page 5, Line 172).
- Page 4, Line 127: Expand CAD.
- Page 9, Line 317: Start the sentence with capital letter. “The beneficial…”
- Citation of references in the text: A consistent pattern should be used.
Page 4, Line 152, 153; Page 5, Line 190; Page 9, Line 337; Page 12, Line 430: The inserted references are in superscript.
Page 11, Line 392, 393, 396, and 399: Use a consistent pattern when inserting multiple references.
- Describe in text HF, MI before using acronym. This will aid wide variety of readers.
- Figures: The font type and size look inconsistent across the figures. The use of a consistent pattern will improve the presentability of the manuscript. For clarity, the upregulation and downregulation can be shown with a different arrow (type/color) than the connection arrows.
Author Response
We greatly appreciate the reviewer’s expert evaluation and suggestions improving our manuscript. We have made the following revisions to strengthen our study.
Q1. The reviewer provides relevant literature for us to refer to and cite.
Response1: We have included the provided references in the revised manuscript.
Q2. The perspective section (page 12, Line 453 onwards) needs to be modified accordingly.
Response2: We have rewritten the phrase to clarify the delivered points.
Q3. a) Page 3, Line 84: Expand ox-LDL as it is mentioned for the first time in the text. Later use acronym (Page 6, Line 214).
- b) Page 3, Line 91: The sentence needs reframing for clarity. Suggestion – “... effect via upregulation of…”
- c) Page 3, Line 96: It should be sequence similarity (54%).
- d) Page 4, Line 121, 124: The references are required when referring to other studies.
- e) Page 4, Line 124: Expand VSMC here and later use as acronym (Page 5, Line 172).
- f) Page 4, Line 127: Expand CAD.
- g) Page 9, Line 317: Start the sentence with capital letter. “The beneficial…”
- h) Citation of references in the text: A consistent pattern should be used.
Page 4, Line 152, 153; Page 5, Line 190; Page 9, Line 337; Page 12, Line 430: The inserted references are in superscript.
Page 11, Line 392, 393, 396, and 399: Use a consistent pattern when inserting multiple references.
- i) Describe in text HF, MI before using acronym. This will aid wide variety of readers.
- j) Figures: The font type and size look inconsistent across the figures. The use of a consistent pattern will improve the presentability of the manuscript. For clarity, the upregulation and downregulation can be shown with a different arrow (type/color) than the connection arrows.
Response3: Thank you for noting these errors, which have been corrected in the revision.

Reviewer 3 Report
This is a well-written review of CTRPs, their function and their roles in human health as well as a very pertinent review of potential interactions with symptoms of COVID-19. I find the text to be concise, well-cited, and the arguments very clear.
Due to the extreme value of this topic and the clarity with which these authors have addressed these topics, I recommend accepting this study as is and getting this text out into the hands of clinicians.
Author Response
Thank you very much for your positive comments and recognition of our work.